

# 2D dipping dike magnetic data interpretation using a robust particle swarm optimization

Khalid S. Essa[1], Mahmoud El-Hussein[1*]

[1] Geophysics Department, Faculty of Science, Cairo University, Giza, P.O. 12613, Egypt.

*Correspondence to:* Mahmoud Elhussein (mahmoudelnouishy@yahoo.com)

**Abstract.** A robust Particle Swarm Optimization (PSO) investigation for magnetic data by a 2D dipping dike has been presented. The interpretive model parameters are: the amplitude coefficient ($K$), the depth to the top of the dipping dike ($z$), exact origin of the dipping dike ($x_0$), and the width of dipping dike ($w$). The inversion procedure is actualized to gauge the parameters of a 2D dipping dike structures where it has been confirmed first on synthetic models without

and with different level of random noise. The results of the inversion demonstrate that the parameters derived from the inversion concur well with the true ones. The root mean square (RMS) is figured by the strategy which is considered as the misfit between the measured and computed anomalies. The technique has been warily and effectively applied to real data examples from China and UK with the presence of ore bodies. The present technique can be applicable for mineral exploration and ore bodies of dike-like structure embedded in the shallow and deeper

subsurface.

## 1 Introduction

These days, tremendous measure of potential field data, for example, magnetic, gravity and SP data are gathered for the environmental and geological applications, including mineral, ores, oil exploration and groundwater investigations

(Reford and Sumner, 1964; Eppelbaum and Khesin, 2012; Essa et al., 2008; Holden et al., 2008; Paoletti et al., 2013; Mehanee, 2014a, Mehanee, 2014b; Pei et al., 2014; Eppelbaum, 2015; Mehanee, 2015; Mehanee and Essa, 2015; Abo-Ezz and Essa, 2016; Essa and Elhussein, 2016a; Maineult, 2016). The intrigued data here are the magnetic data gathered over dikes which considered as the most well-known geologic features and mineral carriers (Abdelrahman and Essa, 2005; Abdelrahman and Essa, 2015; Essa and Elhussein, 2016b; Al-Garni, 2015).

The estimation of model parameters ($z$, $\alpha$, $K$, $w$, and $x_0$) of a buried 2D dipping dike structure is an essential target in interpretation of magnetic data. Thus, many published methods have presented for interpreting magnetic data, for example, the graphical method (Peters, 1949; Koulomzine et al., 1970; Rao and Murthy, 1978), Curve matching techniques (Hutchison, 1958; McGrath and Hood, 1970; Dondurur and Pamukcu, 2003), Fourier transform method (Bhimasankaram et al., 1978), correlation factors and integration nomograms (Atchuta Rao and Ram Babu, 1981;

Kara et al., 1996; Kara, 1997), the midpoint method (Murty, 1985), Hilbert transform method (Mohan et al., 1982;



Ram Babu and Atchuta Rao, 1991), Euler deconvolution method (Reid et al., 1990), Gauss- Newton method (Won, 1981), complex gradient method (Atchuta Rao et al., 1981), relation diagrams (Ram Babu et al., 1982), the gradient methods (Rao et al., 1973; Abdelrahman et al., 2007; Essa and Elhussein, 2016b), damped least-square ridge regression (Johnson, 1969), Spectral analysis methods (Bhattacharya, 1971; Sengupta and Das, 1975; Cassano and

Rocca, 1975), modular neural network inversion (Al-Garni, 2015),  an automated numerical method (Keating and Pilkington, 1990), a new semi-automatic technique (Cooper, 2012), a non-linear constrained inversion technique (Beiki and Pedersen, 2012). However, the drawbacks of these methods are that they are highly subjective where they can lead to substantial errors in parameter estimations, rely upon trial and error till achieve the best fit between the measured and computed anomaly, require initial starting models which are close enough to the true solution, depends

on the precision of separation of regional and residual magnetic anomalies from the measured magnetic anomaly, either the dip of the dike or inclination of magnetic field is assumed for the guarantee of a good overall performance.

In this study, PSO is utilized for the magnetic anomalies due to 2D dipping dike to calculate the interpretive model parameters ($z$, $\alpha$, K, w, and $x_0$). Our new approach was tested on synthetic data without and with different level of Gaussian noise (5%, 10%, 15%, and 20%).  Here, we apply the new calculation on magnetic data by utilizing two

field cases from China and United Kingdom. So as to show the benefits the proposed design algorithm, the outcomes obtained are compared against borehole drilling information.

**2 The method**

The magnetic anomaly by a 2D dipping dike structure is given by (Hood, 1964; McGrath and Hood, 1970):

$T(x_i, z, d, \alpha) =$

$$K\left[\sin\alpha\left(\tan^{-1}\left(\frac{x_i+d}{z}\right) - \tan^{-1}\left(\frac{x_i-d}{z}\right)\right) - \frac{\cos\alpha}{2}\ln\left(\frac{(x_i+d)^2+z^2}{(x_i-d)^2+z^2}\right)\right],$$

$$i = 1,2,3,4,\ldots\ldots\ldots. N \qquad (1)$$

where $z$ is the depth (m) to the top, $d$ is the half-width (m), $\alpha$ (deg.) is the index angle and K (mA/m) is the amplitude coefficient.

Kennedy and Eberhart, (1995) created a PSO-algorithm. It depends on the reproduction of the obvious social conduct of birds, fishes and insects in nourishment searching. PSO-calculation effectively connected and applied in many disciplines, like model development (Cedeno and Agrafiotis, 2003), biomedical pictures (Wachowiak et al., 2004), electromagnetic optimizations (Boeringer and Werner, 2004), hydrological issues (Chau, 2008), and different geophysical application (Alvarez et al., 2006; Shaw and Srivastava, 2007). In this calculation the birds representing

the particles or models, every particle has an area vector which represent the parameters value and a speed vector. For instance, for a five dimensional optimization issue, every particle or individual will have an area in a five dimensional space which represent an answer (Eberhart and Shi, 2001). Every particle changes its position at every



progression of the operation of the calculation, this position is updated during the iteration process considering the best position came to the particle which is known as the $T_{best}$ model and the best area acquired any particle in the group which is known as the $J_{best}$ model, this update is described in equation 2 and 3 (Sweilam et al., 2007)

$$V_i^{k+1} = c_3 V_i^k + c_1 \mathrm{rand}()\left(T_{best} - P_i^{k+1}\right) + c_2 \mathrm{rand}()\left[\left(J_{best} - P_i^{k+1}\right)P_i^{k+1}\right] = P_i^k + V_i^{k+1}, \quad (2)$$

$$x_i^{k+1} = x_i^k + v_i^{k+1}. \qquad (3)$$

where $v_i^k$ is the speed of the particle i at the $k^{th}$ iteration, $P_i^k$ is the current position of the ith particle at the $k^{th}$ iteration, rand() is a random number between 0 and 1, $c_1$ and $c_2$ are positive constant numbers known as cognitive coefficient and social coefficient, respectively, which control the individual and the social conduct, they are normally taken as 2 (Sweilam et al., 2007) yet some late researches provide that choosing $c_1$ greater than $c_2$ but $c1+c2 \leq 4$ may give better results (Parsopoulos and Vrahatis, 2002), $c_3$ is the inertial coefficient which control the speed of the particle, since the

large values may make the particles miss up the good solutions and the small values may result in not enough search space for investigation (Sweilam et al., 2007), its typically taken less than 1, $x_i^k$ is the position of the particle i at the $k^{th}$ iteration.

The five controlled-model parameters (z, K, $\theta$, $x_o$, and w) can be estimated by applying the PSO-algorithm on the following objective function (Santos, 2010)


$$Q = \frac{2\sum_{i=1}^{N}|T_i^m - T_i^c|}{\sum_{i=1}^{N}|T_i^m - T_i^c| + \sum_{i=1}^{N}|T_i^m + T_i^c|}, \qquad (4)$$

where N is the number of reading points, $T_i^m$ is the measured magnetic anomaly at the point $x_i$, $T_i^c$ is the calculated magnetic anomaly at the point $x_i$.

The magnetic anomaly from equation 1 is calculated at each iterative step for each $x_i$ using the PSO-algorithm, Figure 1 represents the work flow of the PSO-algorithm, the RMS between the observed and calculated data is computed from the following formula:

$$RMS = \sqrt{\frac{\sum_{i=1}^{N}[T(x_i) - T_c(x_i)]^2}{N}}, \qquad (5)$$

where $T(x_i)$ is the measured field and $T_c(x_i)$ is the computed field. This is considered as the misfit between the

observed and calculated anomalies.





## 3 Results

In this paper, noisy free and contaminated data with different level of random noise (5, 10, 15, 20%) are utilized to exhibit the execution and appropriateness of the proposed technique.

### 3.1 Noise free data

The PSO-technique was applied to synthetic magnetic anomaly by a dipping dike with the following parameters; K =
795.78 mA/m, z = 8 m, α = 40°, d = 3, $x_o$ = 0 m, and profile length = 80 m. In this situation, there is no noise in the data, so we begin test our procedure utilizing 100 models. The best model was come after 700 iterations and the ranges of the parameters are appeared in Table 1. The inverted-parameters that controlled the body dimensions have a good correspondence with the theoretical values (Table 1) which the percentage errors in the inverted-model parameters equal zero.

### 3.2 Contaminated synthetic data

Noisy-data considered as a critical part in geophysics. With a specific end goal to investigate the conduct of noise corrupted data.

At the principal, we forced 5% of arbitrary Gaussian noise on the magnetic data of the dipping dike model (Figure 2). The inverted-parameters (K, z, α, and d) are given in Table 1. Table 1 demonstrates that the rate of error in the inverted-
model parameters are 10%, 1.25%, 1.725%, 16.67%, respectively, and the RMs error is 3.49 mA/m.

Besides, we imposed 10% of arbitrary Gaussian noise on the same synthetic anomaly (Figure 3). The inverted-parameters (K, z, α, and d) are given in Table 1. Table 1 shows that the percentage of error in the inverted-model parameters are 13%, 1.25%, 0.9%, 20%, respectively, and the RMs error is 7.32 mA/m.

Thirdly, we embed 15% of random Gaussian noise on a similar synthetic anomaly (Figure 4). The inverted-parameters
(K, z, α, and d) are given in Table 1. Table 1 shows that the rate of error in the inverted-model parameters are 16.5%, 0%, 1.35%, 23.33%, respectively, and the RMs error is 13.59 mA/m.

At last, we imposed 20% of random Gaussian noise on a similar synthetic anomaly (Figure 5). The inverted-parameters (K, z, α, and d) are given in Table 1. Table 1 shows that the percentage of error in the inverted-model parameters are 20%, 2.5%, 1.2%, 16.67%, respectively, and the RMs error is 15.68 mA/m.

From the above outcomes, the estimations of the inverted-model parameters (K, z, α, d and $x_o$) for the synthetic example without and with various level of noise are in good correspondence with the true-values.




## 4 Field examples

Keeping in mind the end goal to inspect the productivity and the legitimacy of the suggested method, we have connected our new calculation to two real field cases with expanding complexity of the geological models taken from

the published literature.

### 4.1 Magnetic anomaly of Magnetite Iron Deposit, China

Figure 6 demonstrates the measured magnetic anomaly profile M163-1 in the magnetite iron deposit, Western Gansu Province, China (Guo et al., 1998) of length 222.5 m and the sampling interval is 2.78 m. This magnetic field anomaly is interpreted by the proposed method. Table 2 shows the ranges and results of estimated parameters (K = 8116.91

mA/m, z = 22.24 m, α = 57.99º, d = 9.174, $x_o$ = 0.01 m). The evaluated model has been computed and contrasted with the true ones (Figure 6). The final outcomes computed are contrasted with the borehole drilling information (z = 20-25 m and d = 9 m) as far as various parameters (Table 3).

### 4.2 Magnetic anomaly in Red Hill Farm, UK

In this part, we will utilize a PSO algorithm to evaluate the interpretive model parameters of magnetic anomaly profile

taken in south of Red Hill Farm, Millom, United Kingdom (Hallimond and Whetton, 1939). The length of the profile is 61 m and the digitizing interval is 0.76 m (Figure 7). This magnetic field anomaly is interpreted by the proposed method. Table 4 demonstrates the ranges and outcomes of estimated parameters (K = 175.07mA/m, z = 19 m, α = 75.6º, d = 2.28 m, $x_o$ = -0.1). The estimated model has been computed and contrasted with the true ones (Figure 7) There is a good correlation between results assessed from our technique and the borehole drilling information

published in Hallimond and Whetton (1939) (where z = 18.29 m and d = 2.29 m).

At last, we assess the adequacy and proficiency of our technique, we additionally contrast its execution with some state of art algorithms. Tables 2 and 4 condense the inverted results. As can be found in these tables, surprisingly, our inversion algorithm significantly outperforms the up-to-date model parameters estimation for two field examples. For all field example, our inversion gives a full picture of the model parameters rather than other methods which did not

give a fully interpretation.






## 5 Conclusions

This paper illustrated the utilization of new algorithm in assessing the model parameters ($z$, $x_o$, $d$, $\alpha$ and $K$) of a 2D
dipping dike model. The viability of the suggested algorithm is exhibited on five troublesome examples including
noisy free data, contaminated data, and two real field data. The new approach has the ability to get the better quality
solution, and has better convergence characteristics and computational efficiency. The comparison of the results with
drilling information reported in the literature demonstrated the prevalence of the suggested method and its potential
for solving magnetic problem. From the outcomes obtained, it is finished up new inversion algorithm is a promising
technique for solving the quantitative interpretation of magnetic data. In the future work, we will attempt to propose
some enhanced version of this new inversion algorithm to solve the problem.

## 6 Data availability

First our algorithm is applied on synthetic data created from equation 1 with and without Gaussian noise (Hood,
1964; McGrath and Hood, 1970), the results were given in section 3. Then the proposed algorithm applied to two
field examples the first one is profile over the magnetite iron deposit, Western Gansu Province, China (Guo et al.,
1998) of length 222.5 m (Figure 6) and the sampling interval is 2.78 m, while the second one was magnetic anomaly
profile taken in south of Red Hill Farm, Millom, United Kingdom (Hallimond and Whetton, 1939). The profile
length is 61 m and the sampling interval is 0.76 m (Figure 7).

### Acknowledgements

Authors would like to thank Prof. Jothiram Vivekanandan, Chief-Executive Editor, and Prof. Lev Eppelbaum, the
Associate Editor for his constructive comments for improving our manuscript.





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



**Table 1. Numerical results for a 2D dipping dike model without and with various level of 5%, 10%, 15%, and 20% of Gaussian noise (K = 795.78 mA/m, z = 7 m, α = 40 °, d = 3, $x_o$ = 0 m, and profile length = 80 m).**

| Type of body | Parameters | Used ranges | Result | e (%) | RMS (mA/m) |
|---|---|---|---|---|---|
| | | without random Gaussian noise | | | |
| | K (mA/m) | 397.89-1591.55 | 795.78 | 0 | |
| | z (m) | 1-12 | 8 | 0 | |
| | α (°) | 10-60 | 40 | 0 | 0 |
| | d (m) | 1-8 | 3 | 0 | |
| | $x_o$ (m) | -80-30 | 0 | 0 | |
| | | with 5% random Gaussian noise | | | |
| | K (mA/m) | 397.89-1591.55 | 875.35 | 10 | |
| | z (m) | 1-12 | 8.1 | 1.25 | |
| | α (°) | 10-60 | 39.31 | 1.725 | 3.49 |
| | d (m) | 1-8 | 2.5 | 16.67 | |
| | $x_o$ (m) | -80-30 | 0.11 | ------ | |
| | | with 10% random Gaussian noise | | | |
| | K (mA/m) | 397.89-1591.55 | 899.23 | 13 | |
| | z (m) | 1-12 | 8.1 | 1.25 | |
| Dipping dike | α (°) | 10-60 | 40.36 | 0.9 | 7.32 |
| | d (m) | 1-8 | 2.4 | 20 | |
| | $x_o$ (m) | -80-30 | -0.08 | ------ | |
| | | With 15% random Gaussian noise | | | |
| | K (mA/m) | 397.89-1591.55 | 927.08 | 16.5 | |
| | z (m) | 1-12 | 8 | 0 | |
| | α (°) | 10-60 | 39.46 | 1.35 | 13.59 |
| | d (m) | 1-8 | 3.7 | 23.33 | |
| | $x_o$ (m) | -80-30 | 0.10 | ------ | |
| | | With 20% random Gaussian noise | | | |
| | K (mA/m) | 397.89-1591.55 | 954.93 | 20 | |
| | z (m) | 1-12 | 7.8 | 2.5 | |
| | α (°) | 10-60 | 40.48 | 1.2 | 15.68 |
| | d (m) | 1-8 | 2.5 | 16.67 | |
| | $x_o$ (m) | -80-30 | -0.25 | ------ | |




**Table 2. Numerical results for the magnetite iron deposit field example, China.**


| Parameters | Used Ranges | Result | RMS (mA/m) |
|---|---|---|---|
| K ( mA/m) | 79.58-1591.55 | 8116.91 | |
| z (m) | 16-28 | 22.24 | |
| α (°) | 20-80 | 57.99 | 198.73 |
| d (m) | 5-12 | 9.174 | |
| $x_o$ (m) | -10-10 | 0.01 | |

**Table 3. Correlation between numerical results obtained from drilling data and our method for magnetite iron deposit field example, China.**

| Method \ Parameters | Drilling data, Guo, et al. (1998) | Present method |
|---|---|---|
| K ( mA/m) | ------- | 8116.91 |
| z (m) | 22-25 | 22.24 |
| α (°) | ------- | 57.99 |
| d (m) | 9 | 9.174 |
| $x_o$ (m) | -------- | 0.01 |




**Table 4. Numerical results for magnetic anomaly in Red Hill Farm, UK field example.**

| Parameters | Used Ranges | Result | RMS (mA/m) |
|---|---|---|---|
| K (mA/m) | 79.58-795.78 | 175.07 | |
| z (m) | 20-30 | 19 | |
| $\alpha$ (°) | 20-90 | 75.6 | 0.74 |
| d (m) | 1-10 | 2.28 | |
| $x_o$ (m) | -10-10 | 0.1 | |













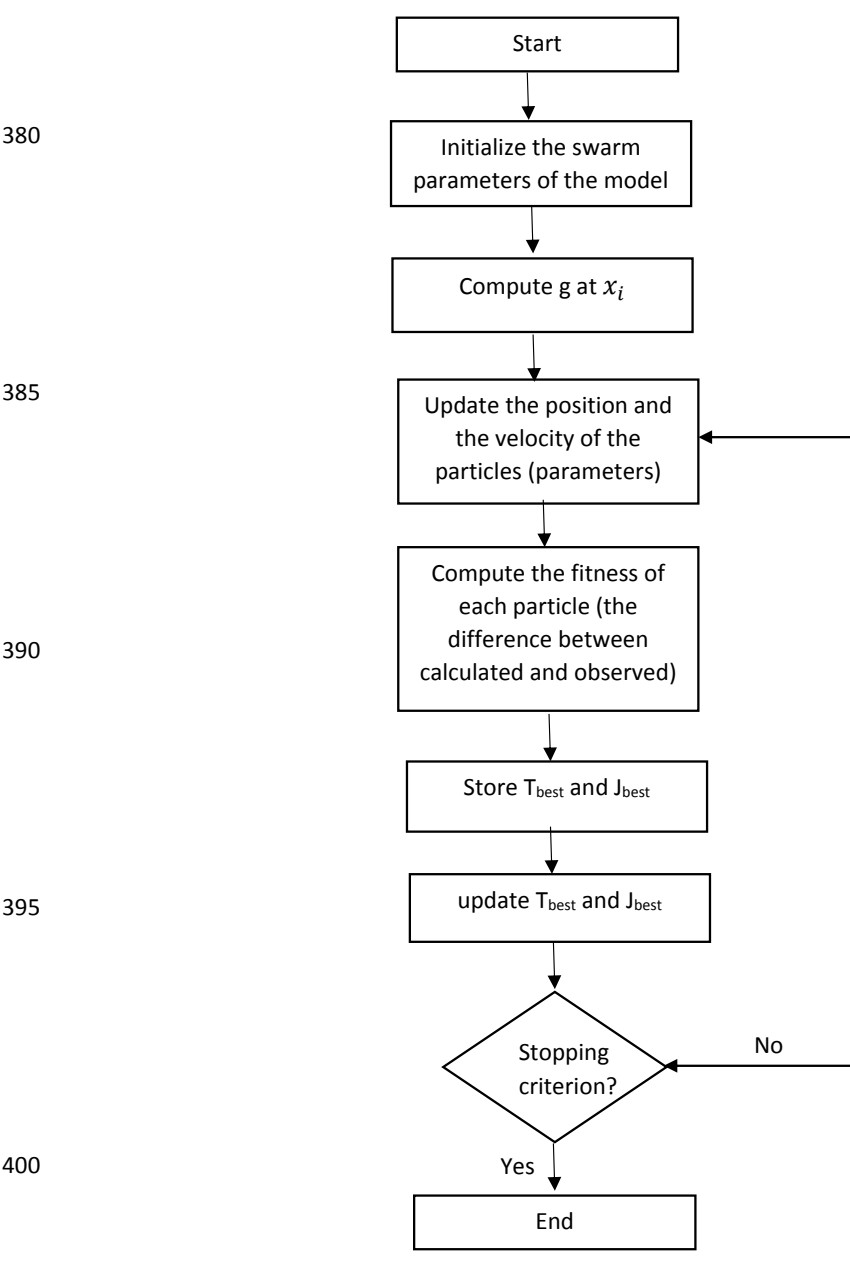

**Figure 1: work-flow of PSO-algorithm applied to magnetic anomalies interpretation**









**Figure 2: Total magnetic anomaly of a buried dipping dike-like a geologic structure with K = 795.78 mA/m, z = 8 m, α = 40º, d= 3, $x_o$ = 0 m, and profile length = 80 m, without and with 5% random Gaussian noise.**



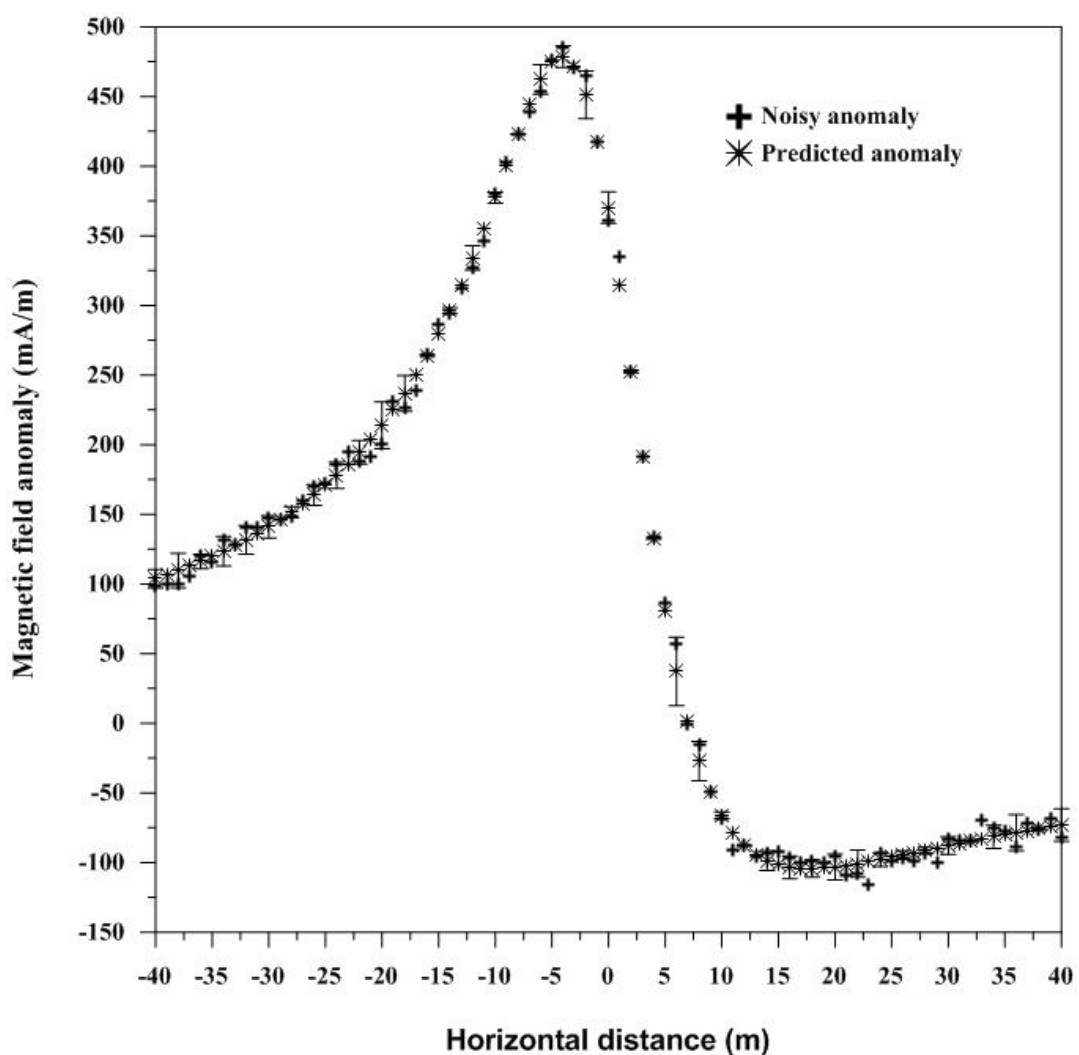


**Figure 3: Total magnetic anomaly of a buried dipping dike-like a geologic structure with K = 795.78 mA/m, z = 8 m, α = 40°, d= 3, $x_o$ = 0 m, and profile length = 80 m, without and with 10% random Gaussian noise.**




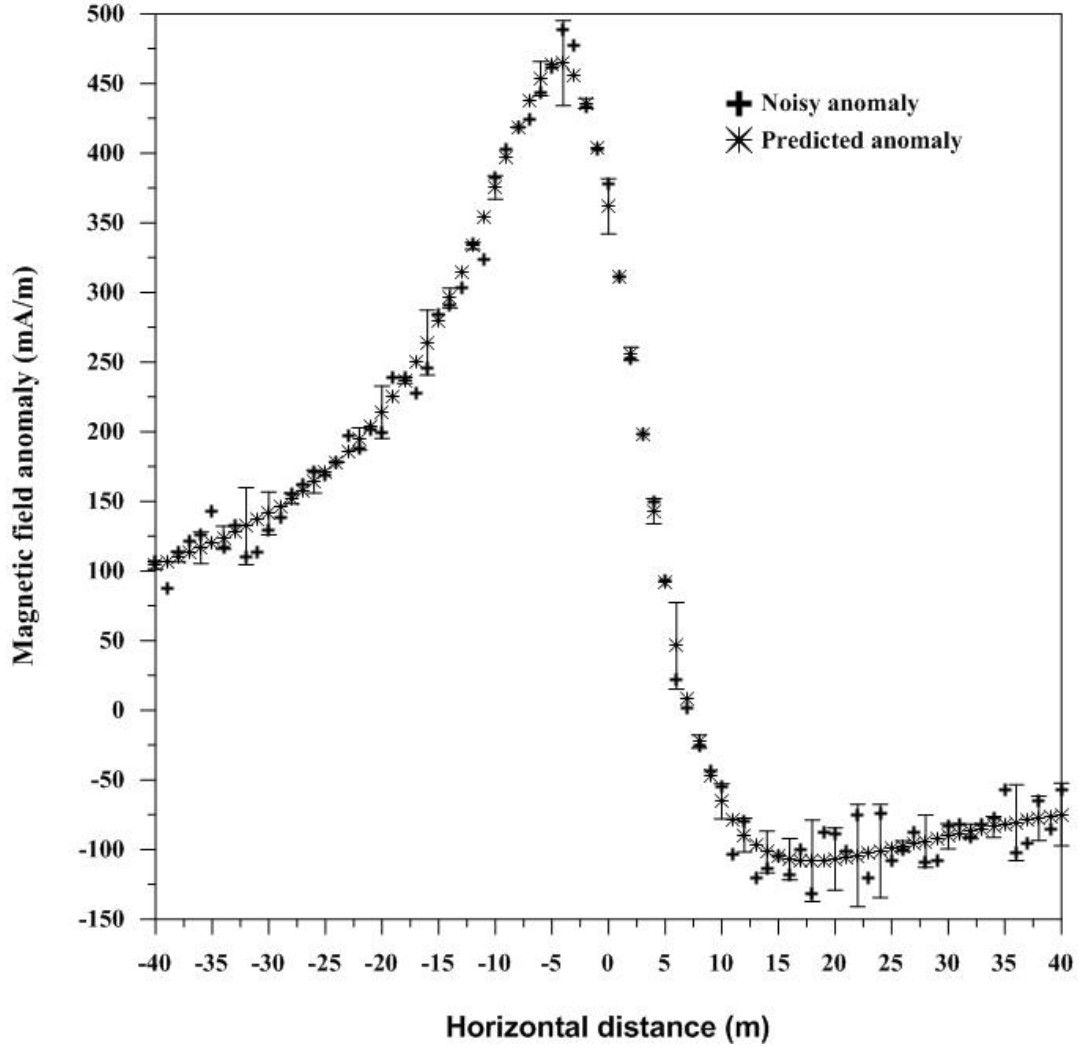

**Figure 4: Total magnetic anomaly of a buried dipping dike-like a geologic structure with K = 795.78 mA/m, z = 8 m, α = 40°, d= 3, $x_o$ = 0 m, and profile length = 80 m, without and with 15% random Gaussian noise.**




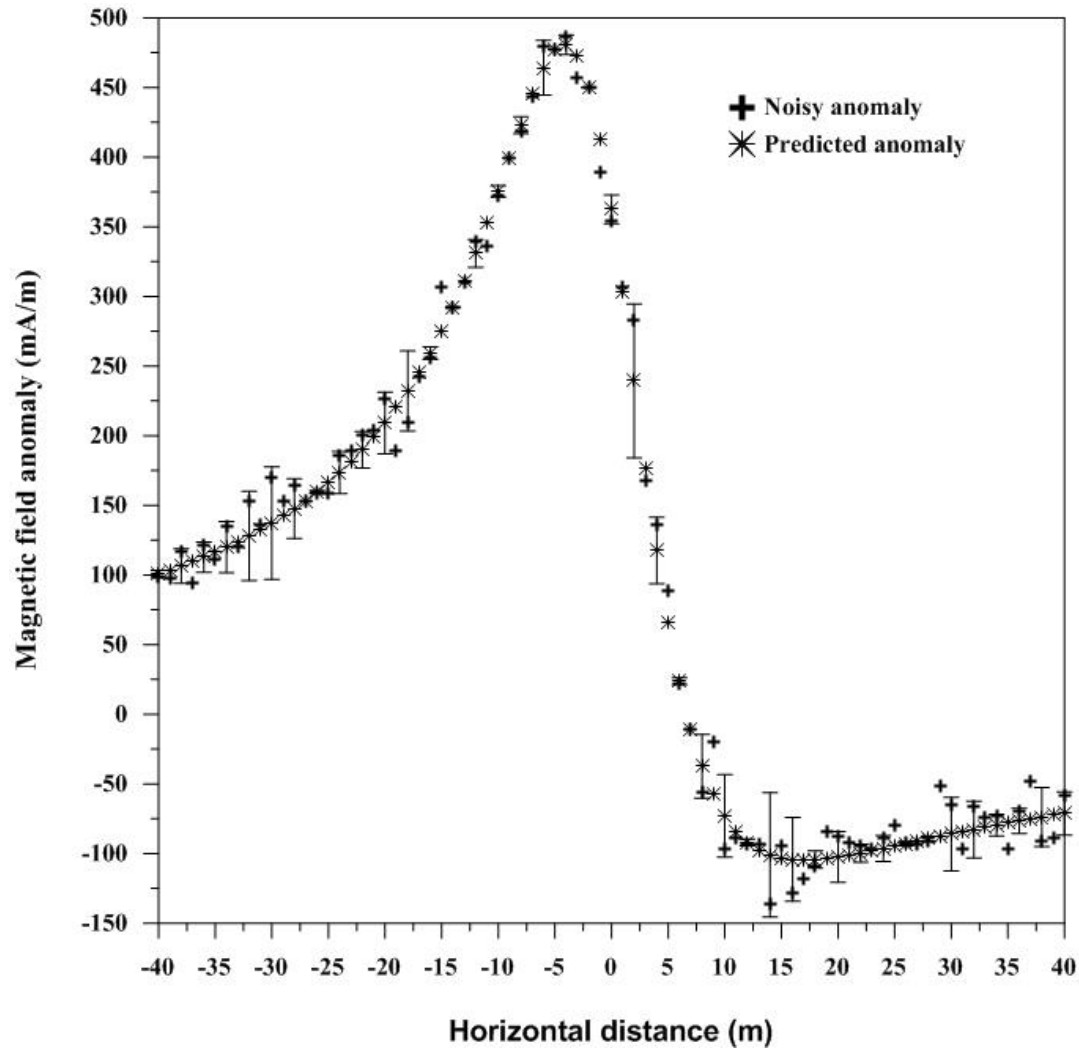

**Figure 5: Total magnetic anomaly of a buried dipping dike-like a geologic structure with K = 795.78 mA/m, z = 8 m, α = 40°, d= 3, $x_o$ = 0 m, and profile length = 80 m, without and with 20% random Gaussian noise.**










**Figure 6: A total magnetic anomaly profile over a magnetite iron deposit, Western Gansu Province, China (open circle) and the estimated magnetic anomaly (black circle) using PSO-algorithm.**


**Figure 7: A vertical magnetic anomaly profile over southern Red Hill Farm, Millom, United Kingdom (open circle) and the estimated magnetic anomaly (black circle) using PSO-algorithm.**