# Peer review of "2D dipping dike magnetic data interpretation using a robust particle swarm optimization"

_Geoscientific Instrumentation, Methods and Data Systems, 2017_

## Author Comment (AC1) · 19 Sep 2017

Dear Sirs, Good evening. Thank you so much for your help in this matters. Best regards Mahmoud
* * *

---

## Referee Comment (RC3) · Anonymous Referee #2 · 5 Apr 2018

Dear authors,

1. why do you use the objective function of expression (4), which is written as the absolute value formation? Usually, the l2 normal objective function is used in potential field data inversion.

2. Unit of magnetic anomaly is nT (not mA/m). Please make clear the magnetic induction intensity, magnetic field strength, magnetization and susceptibility. Magnetization usually is written as M (not K).

3. Please show the inverted dike models in Figure 6, Figure 7. Also show the true and inverted models in figures of synthetic examples.

[Figure]

4. The symbols of observed data and predicted data are not clear to distinguish. Please use color symbols or lines.

5. Title "Parametric inversion of magnetic model for 2D dipping dike model using particle swarm optimization".

6- Could you investigate the effect of a regional background field on the inversion results?

Kind Regards,
* * *

---

## Editor Comment (EC1) · J. Vivekanandan (Editor) · 22 Jun 2018

Dear Dr. Mahmoud Elhussein, Thank you for submitting the manuscript entitled 'Parametric inversion of magnetic model for 2D dipping dike model using a robust particle swarm optimization' for consideration of publication in Geoscientific Instrumentation, Methods and Data Systems journal. The associate editor is unable to secure adequate reviews of the manuscript for a disposition. The journal has contacted a large number of reviewers none of which has agreed to do the review. This may be due to the challenges the prospective reviewers have found in the technical and scientific contents of your manuscript.

[Figure]

Since the journal is unable to secure adequate reviews of your manuscript, we return the manuscript. No decision is made on the manuscript. Please note it is not a rejection of your manuscript.

Also, I would like to point out potential conflict of interest description at https://www.geoscientific-instrumentation-methods-and-data-systems.net/about/competing_interests_policy.html

Please let me know if you need any additional information from me.

Regards, Vivek.

———————————————————